# The Association between Vitamin D, Interleukin-4, and Interleukin-10 Levels and CD23+ Expression with Bronchial Asthma in Stunted Children

**DOI:** 10.3390/biomedicines11092542

**Published:** 2023-09-15

**Authors:** Gartika Sapartini, Gary W. K. Wong, Agnes Rengga Indrati, Cissy B. Kartasasmita, Budi Setiabudiawan

**Affiliations:** 1Division of Allergy Immunology, Department of Child Health, Doctoral Study Program, Faculty of Medicine Universitas Padjadjaran, Bandung 40161, West Java, Indonesia; 2Department of Paediatrics, Faculty of Medicine, The Chinese University of Hong Kong, Hong Kong, China; wingkinwong@cuhk.edu.hk; 3Department of Clinical Pathology, Faculty of Medicine Universitas Padjadjaran, Hasan Sadikin General Hospital, Bandung 40161, West Java, Indonesia; agnes.indrati@unpad.ac.id; 4Division of Respirology, Department of Child Health, Faculty of Medicine Universitas Padjadjaran, Hasan Sadikin General Hospital, Bandung 40161, West Java, Indonesia; cbkarta@gmail.com; 5Division of Allergy Immunology, Department of Child Health, Faculty of Medicine Universitas Padjadjaran, Hasan Sadikin General Hospital, Bandung 40161, West Java, Indonesia; budi.setiabudiawan@unpad.ac.id

**Keywords:** 25(OH)D, bronchial asthma, CD23+, IL-4, IL-10, stunted

## Abstract

Children with stunted growth have an increased risk of wheezing, and studies have shown that low levels of vitamin D and interleukin (IL)-10, along with increased IL-4 levels and CD23+ expression, are present in stunted and asthmatic children. To date, it is not known whether these factors are related to the incidence of asthma in stunted children. This case-control study investigated the association between vitamin D, IL-4, and IL-10 levels and CD23+ expression with bronchial asthma in stunted children. The study included 99 children aged 24–59 months, i.e., 37 stunted-sthmatic children (cases), 38 stunted children without asthma, and 24 non-stunted children with asthma. All children were tested for their 25(OH)D levels using chemiluminescent immunoassay (CLIA), IL-4 and IL-10 levels were measured through enzyme-linked immunosorbent assay (ELISA) testing, and CD23+ expression was measured through flow cytometry bead testing. The data were analyzed using chi-squared, Kruskal-Wallis, and Mann-Whitney tests. The results showed that stunted asthmatic children had a higher incidence of atopic family members than those without asthma. Additionally, stunted asthmatic children had a higher prevalence of vitamin D deficiency (48.6%) than the control group (44.7% and 20.8%). Furthermore, stunted asthmatic children had significantly lower levels of 25(OH)D [20.55 (16.18–25.55), *p* = 0.042] and higher levels of IL-4 [1.41 (0.95–2.40), *p* = 0.038], although there were no significant differences in IL-10 levels and CD23+ expression. The study concluded that low vitamin D and high IL-4 levels are associated with bronchial asthma in stunted children, while IL-10 and CD23+ do not show a significant association.

## 1. Introduction

Stunting is an ongoing issue worldwide, particularly in low- and middle-income countries. According to the United Nations International Children’s Emergency Fund (UNICEF), World Health Organization (WHO), and the World Bank Group, an estimated 148.1 million children under five years old, or 22.3%, were affected globally in 2022 [1]. In lower-middle-income countries, the prevalence of stunting is 29.1%, as reported by the World Bank. In Indonesia, West Java has the 18th highest proportion of stunted children, at 31.1%, higher than the national average [2]. The second Sustainable Development Goal (SDG) aims to end all forms of malnutrition by reducing the number of stunted children under the age of 5 by 40% by 2025 [3].

Stunting is primarily caused by chronic malnutrition and infections. According to Hawlader et al., stunted children living in rural Bangladesh are at a higher risk of wheezing. Malnourished children often have less body fat, resulting in poor lung growth, reduced lung function, and increased asthma symptoms [4]. Mokhtar et al. reported that vitamin D deficiency in Ecuador was more common among stunted and underweight children aged 6 to 36 months [5]. Stunting is associated with lower levels of 25(OH)D, which can increase the incidence of asthma [5].

Asthma is a heterogeneous disease characterized by chronic airway inflammation with respiratory symptoms such as wheezing, shortness of breath, chest tightness, and cough, which vary in time and intensity, along with variable expiratory airflow limitation [6]. To date, the main factors associated with the occurrence of asthma and allergies have been identified, including exposure to allergens and pollutants, genetics, changes in diet, and nutritional status [7]. Deficiencies of certain nutrients, such as vitamin D, could potentially be a cause of asthma and allergies. Vitamin D is linked to the immune system and intrauterine lung development; a higher vitamin D intake in pregnant women reduces the risk of asthma by 40% in children aged 3–5 years [8]. Vitamin D deficiency is strongly associated with asthma, allergic rhinitis, and wheezing. Children with vitamin D deficiency are at 6.3-fold higher risk of developing asthma than children with normal vitamin D levels. This is because vitamin D exerts immunomodulatory effects that inhibit T helper 1 (Th1) cell activation, modulate Th2 cells, and increase regulatory T cell (Treg) activity. Asthma is considered to be a Th2-mediated disease [9,10].

Studies have shown that mice with a deficiency in vitamin D exhibit reduced levels of Treg cells that secrete regulatory cytokines such as IL-10 and TGF-β, resulting in an increase in Th2 cells [11]. Interleukin-10 is linked to the prevention of asthma due to its immunosuppressive and anti-inflammatory properties. Atopic asthma patients generally have lower levels of IL-10 compared with healthy individuals. The positive impact of 1,25(OH)2D3 on Treg cells and IL-10, which help suppress Th2 responses, may be one of the reasons why vitamin D is beneficial for asthma treatment [12,13].

The development of asthma is linked to an excess production of Th2 cytokines, specifically IL-4 and IL-5 [14]. Th2 cells generate IL-4, which is crucial in promoting proinflammatory functions that contribute to asthma. These functions include the differentiation of T cells into Th2 and inducing the class switching of B cells to produce IgE [15,16]. Plasma cells produce immunoglobulin E, which binds to the high-affinity IgE receptor (FceRI) in mast cells. When exposed to allergens, the cross-linking of IgE leads to the release of histamine, leukotrienes, and prostaglandins, resulting in bronchoconstriction [15]. A study by Al-Daghri et al. revealed that children with asthma had significantly higher levels of serum IL-4 and IgE compared with controls [17].

IgE synthesis is facilitated by interleukin-4 [18]. In addition to binding to the high-affinity IgE receptor (FcεRI), IgE binds to the low-affinity IgE receptor (CD23+, FcεRII). This receptor plays a crucial role in the allergic inflammatory process. In particular, CD23+ (FcεRII) is a key molecule found on the surface of B lymphocytes; it enables IgE-facilitated allergen presentation and the subsequent activation of allergen-specific T cells [19]. Gagro et al. reported that CD23+ expression on lymphocytes increased in allergic asthmatic children and was positively correlated with serum IgE levels [20].

Studies have shown that leptin plays a crucial role in the development of asthma in obese individuals, as high levels of leptin are associated with asthma [21]. However, there has been no research on the role of leptin in asthma among malnourished children. Malnutrition can lead to a decrease in adiposity mass, resulting in lower levels of leptin. Leptin increases Th1 and Th17 cytokines while inhibiting the production of Th2 cytokines and Treg proliferation [22]. Mice with leptin deficiency exhibit an increased production of Th2 cytokines (IL-4 and IL-10) and a decreased secretion of Th1 cytokines (IL- 2, IFN-γ, and TNF-α). This indicates that leptin plays a role in the balance of Th1 and Th2 cytokines. Low leptin levels in stunted children can cause a shift in Th1/Th2 balance towards Th2, resulting in an increase in IL-4 production. Stunted children also experience immune response abnormalities with the increased production of cytokines IL-4 and IL-10 compared with well-nourished children [23,24,25]. They experience impaired T cell responses, as well as an increased proportion of the total number of B cells containing low affinity IgE receptors (CD23+), IL-4 levels, and total IgE [26].

The information provided suggests that vitamin D, IL-4, IL-10, and CD23+ expression could potentially contribute to the development of bronchial asthma in stunted children, as previously explained by Sapartini et al. in Figure 1 [27]. However, no research has been conducted on this topic yet. It is crucial to prevent asthma in stunted children, as the combination of stunting and asthma can have serious health consequences that impact their quality of life. A study investigated the possible association between vitamin D, IL-4, and IL-10 levels and CD23+ expression with bronchial asthma in stunted children.

## 2. Materials and Methods

This study was a case-control study that included children aged 24–59 months who were either stunted or not stunted and either had or did not have asthma. The participants were selected from all Bandung District Health Centers (62 District Health Centers) between October 2021 and October 2022. Prior to the study, approval was obtained from the Ethics Committee of Universitas Padjadjaran with the number 572/UN6.KEP/EC/2021, and informed consent forms were signed by the parents of the participating children.

### 2.1. Data Collection and Sample Study

The study recruited participants through consecutive sampling, selecting those who met the inclusion criteria and did not meet the exclusion criteria. Inclusion criteria are as follows: (1) stunted and asthmatic if stunted children (height for age/HFA < −2 SD with weight for age/WFA > −3 SD to 2 SD according to the WHO child growth standard/WHOCGS curve) had bronchial asthma, (2) stunted but without asthma if stunted children did not have asthma, and (3) non-stunted and asthmatic if children with normal height and weight according to the WHOCGS curve had bronchial asthma. The sample size was determined using a formula to test the difference between two means, with a significant level of 5% (Zα = 1.96) and a power test of 95% (Zβ = 1.65). Each group required a minimum of 20 participants. During the study, the participants had their weight and height measured, and their height was compared with the WHOCGS curve to determine if they were stunted or non-stunted for their age. In addition, the participant’s parents were interviewed using the global initiative for asthma (GINA) and asthma risk factor questionnaire to identify the diagnosis of asthma and any associated risk factors [26]. Based on the results, the participants were divided into three groups: stunted asthmatic (case group), stunted without asthma, and non-stunted asthmatic (control group).

Participants who met the following exclusion criteria were not included in the study: (1) those with major congenital abnormalities or certain syndromes such as Down syndrome; (2) stunted children who were overweight, obese, and severely malnourished (weight for age < −3 SD); (3) children with acute and chronic infections; (4) those who had received vitamin D supplementation; and (5) children with bronchial asthma who had received steroid inhalation therapy for ≥12 weeks with a dose of 200 micrograms or higher.

Peripheral venous blood samples were collected from both the case and control groups to measure the levels of 25(OH)D, IL-4, IL-10, and CD23+ expression. Fresh blood samples were divided into two parts: the first sample was sent to the Immunology Division of the Biomedical Laboratory for examination of CD23+ expression; the second sample was sent to the Clinical Pathology Laboratory of Dr. Hasan Sadikin Hospital Bandung, and then the serum was stored at −80 °C. The pooled blood samples were used to measure the levels of 25(OH)D, IL-4, and IL-10, which were determined simultaneously after all samples were collected. Figure 2 illustrates the selection process for study participants.

### 2.2. Anthropometric Measurement

To measure the weight of children aged ≥2 years who could stand on their own, we used a calibrated digital scale. The scale was placed on a flat and hard surface. We used a stadiometer with a fixed vertical backboard and an adjustable headpiece to take height measurements. For children aged ≥2 years who could stand independently without assistance, we assessed their maximum vertical size, which is their height. Stunting was identified when height < −2 SD from the median WHOCGS curve for the same age and sex [1].

### 2.3. Vitamin D Level Measurement

The level of 25(OH)D is a measure of the inactive form of vitamin D, which is produced by the liver when it metabolizes vitamins D2 and D3 through the 25-hydroxylase enzyme. Serum 25OHD was measured using the Architect 25-OH Vitamin D assay, CLIA that determines the level of 25-hydroxyvitamin D (25-OH vitamin D) in human serum and plasma. The manufacturer’s instructions were strictly followed, and the reagent was purchased from Longford Ireland. An Architect i2000SR immunoassay analyzer (Abbott Laboratories, Longford, Ireland) was used to measure the serum 25(OH)D concentrations. To ensure accuracy, quality control materials were tested daily using FDA-approved methods to ensure that the measurement value was within the analytical measuring range. When the batch of reagent was changed, a standard control was used to adjust the measuring curve. This study adopted the Endocrine Society’s classification to categorize levels of 25(OH)D as deficient (≤20 ng/mL), sufficient (between 21 and 29 ng/mL), and sufficient (≥30 ng/mL) [28].

### 2.4. Measurement of IL-4 and IL-10 Levels 

Interleukin-4 is a pro-inflammatory cytokine produced by Th2 cells and plays an important role in bronchial asthma. IL-10 is produced by Treg cells. The ELISA kit was used with the Sandwich-ELISA principle to test both samples. The kit included a micro-ELISA plate that had been pre-coated with an antibody specific to human IL-4 and IL-10, purchased from Elabscience Biotechnology Inc.^®^ (Houston, TX, USA). To conduct the test, samples were added to the micro-ELISA plate wells and combined with a specific antibody. Subsequently, a biotinylated detection antibody specific for human IL-4 and avidin–horseradish peroxidase (HRP) conjugate were added successively to each micro plate well and incubated. Those wells that contained human IL-4, biotinylated detection antibody, and conjugated Avidin–HRP turned blue in color. The enzyme–substrate reaction was terminated by adding a stop solution, causing the color to turn yellow. Finally, the optical density (OD) was measured spectrophotometrically at a wavelength of 450 ± 2 nm. IL-10 was measured in a similar manner.

### 2.5. CD23+ Expression Measurement

CD23+ is a low-affinity receptor for IgE and plays an important role in regulating the IgE response. The measurement of CD23+ expression focused on mCD23+ (CD23+ membrane) expressed by B lymphocytes, using leukocyte and B lymphocyte markers (CD3+ and CD19+, respectively). The assessment of CD23+ expression included the percentage and mean fluorescence intensity (MFI) [29] and was conducted using flow cytometry beads. The reagent used in this study was purchased from San Jose, USA. Flow cytometric analysis was performed on whole blood samples using the specified conjugated antibody. Laser excitation was used at 488 nm, 635 nm, or 540 nm. CD23 expression data were obtained by quantifying peripheral blood samples treated with BD FACS lysing solution using a BD FACScanTM flow cytometer, following the manufacturer’s instructions. The APC-R700 conjugation was read using a red laser (640 nm) with a 685 longpass mirror and a 712/21 bandpass filter. To lyse red blood cells, diluted (1X) BD FACS lysing solution was used as directed. The whole blood was stained with BD Tristest CD3/CD4/CD45 reagent and BD simultest CD3/CD4 reagent. Data analysis was conducted using FlowJo Software 7.5 (TreeStar, Woodburn, OR, USA). Molecule density on cells was calculated only when more than 20 cells of the assessed cell type were positive for CD23+.

### 2.6. Data Analysis

The data were analyzed using both descriptive and analytical methods. Descriptive analysis involved presenting numbers and percentages for categorical data and the mean, standard deviation, median, and interquartile range for numerical data. Analysis involved testing data normality with the Shapiro–Wilk test. The chi-squared statistical test was used to analyze the relationship between two categorical data variables, while the Kruskal–Wallis test was used for numerical data. A post hoc test (Mann–Whitney) was conducted following the Kruskal–Wallis test. A *p*-value of less than 0.05 was considered significant for all tests.

## 3. Results

### 3.1. Participant Characteristics

Out of a total of 99 children, 37 were stunted and asthmatic, 38 were stunted without asthma, and 24 were non-stunted and asthmatic. To determine the impact of confounding variables such as age, sex, exposure to cigarette smoke, and family history of atopic disease on bronchial asthma, a comparison of participant characteristics was conducted. After analyzing their baseline characteristics, it was found that there were no differences in age, sex, weight for height, degree of asthma, allergen exposure, cigarette smoke exposure, delivery method, birth weight, gestational age, and leukocyte count. The results are summarized in Table 1.

The mean age of stunted asthmatic children was older than those who were stunted without asthma. There were more males than females in the stunted asthmatic and non-stunted asthmatic groups. Stunted children with asthma had a higher proportion of family members with atopy (including the father, mother, and siblings) compared with stunted children without asthma and non-stunted children with asthma. To determine which group exhibited a significant difference, a post hoc test was conducted on the family history of atopy. The stunted asthmatic group vs. the stunted without asthma group showed significant differences in the family history of atopy in this study. There was a lower proportion of stunted asthmatic children who were exclusively breastfed than stunted children without asthma and non-stunted asthmatic children (*p* < 0.001).

### 3.2. Vitamin D, IL-4, and IL-10 Levels and CD23+ Expression in Participants

The vitamin D levels, specifically 25(OH)D, were measured and the results are presented in Table 2. The table displays the percentage of participants in each group that are considered deficient, insufficient, and sufficient in terms of vitamin D. The findings show that stunted asthmatic children had a higher proportion of vitamin D deficiency compared with the control group (48.6% vs. 44.7% and 20.8%). Conversely, sufficient vitamin D levels were observed more in non-stunted asthmatic children than in stunted children without asthma and stunted asthmatic children (29.2% vs. 18.4% vs. 10.8%, respectively).

A comparison of vitamin D, IL-4, and IL-10 levels and CD23+ expression was conducted among children who were stunted and asthmatic, stunted without asthma, and non-stunted and asthmatic, as shown in Table 3. The results indicate that there were significant differences in the levels of 25(OH)D and IL-4. Stunted asthmatic children had a deficiency in 25(OH)D levels, while stunted children without asthma and non-stunted asthmatic children had insufficient levels. Moreover, the 25(OH)D levels in stunted asthmatic children were significantly lower than those in stunted children without asthma and non-stunted asthmatic children (*p* = 0.042). IL-4 levels were significantly higher in stunted asthmatic children compared with children who were stunted without asthma and non-stunted asthmatic children (*p* = 0.038). IL-10 levels appeared to be lower in stunted asthmatic children than in the control groups. The ratio of IL-4/IL-10 levels in stunted asthmatic children tended to be higher than those in stunted children without asthma and non-stunted asthmatic children. CD23+ expression (percentage and MFI) in stunted asthmatic children tended to be higher than that in the control groups. In post hoc analysis, it was found that significant differences existed between the group of stunted asthmatic children and non-stunted asthmatic children. Vitamin D levels in stunted asthmatic children were lower than those in non-stunted asthmatic children (*p* = 0.022). However, IL-4 levels in stunted asthmatic children were higher than those in non-stunted asthmatic children (*p* = 0.035). The results of the analysis are reflected in the *p*-values in Figure 3 and Figure 4.

## 4. Discussion

In this study, the associations between vitamin D, IL-4, and IL-10 levels and CD23+ expression with bronchial asthma in stunted children were investigated. This study identified novel discoveries that have not previously been reported. Previous studies have only reported vitamin D, IL-4, and IL-10 levels and CD23+ expression in asthmatic and stunted children separately. The results showed an association between vitamin D and IL-4 levels and bronchial asthma in stunted children, although no association was found between IL-10 levels and CD23+ expression.

The development of asthma is influenced by both genetic and environmental factors [30]. Nikhita and Padmalatha also reported that the risk factor for developing asthma was due to genetic predisposition in the form of a family history of atopy [31]. Di Cicco et al. reported that a family history of atopy is a major risk factor for developing asthma in children [32]. Asthma is a genetically inherited disease that can be passed down from parents to children, with heritability ranging from 35% to 70%. Children whose parents have asthma are almost twice as likely to develop the condition compared with those whose parents do not have it. Additionally, having siblings with a history of asthma can also increase the risk [33]. In this study, new findings were found, namely, the proportion of stunted asthmatic children with a family history of atopy (father, mother, and siblings) was significantly higher than that of non-stunted asthmatic children. In particular, the proportion of biological mothers with atopy was higher than that of biological fathers. Other studies have also shown that having biological parents with atopy increases the likelihood of children developing asthma, with a biological mother presenting a greater risk than a biological father [34]. Therefore, assessing the family history of atopy in stunted children can help identify those at risk of developing asthma.

During early childhood, there is an opportunity to use epigenetic interventions, such as an epigenetic diet, which could prevent or even reverse the negative effects of environmental risk factors in allergic diseases. Human milk is a rich source of microRNAs (miRNAs), which affect the immune system by regulating T and B cell development, dendritic cell differentiation, and the release of inflammatory cytokines. These miRNAs are present in human milk during the first 6 months of breastfeeding [35]. In this study, the proportion of children fed formula milk was higher in stunted asthmatic children compared with stunted children without asthma and non-stunted asthmatic children. Abarca et al. reported that exclusive breastfeeding for at least 6 months can prevent or delay the onset of asthma [36].

Environmental exposure is also a significant factor in the sensitization to environmental allergens and asthma morbidity in children [37]. Increasing evidence shows that exposure to both indoor and outdoor air pollution can contribute to the development of asthma [38]. Infants and young children may develop asthma as a result of exposure to indoor environmental factors [39]. In this study, non-stunted asthmatic children were observed to have more cats from the age of 0–12 months as well as in the last 12 months compared with stunted asthmatic children and stunted children without asthma. However, there is still controversy surrounding the relationship between owning a cat and the onset of asthma. While some researchers, such as Buddha et al., suggest that contact with cats can pose a significant risk factor for asthma, others, such as Taniguchi and Kobayashi, have reported that dog and cat ownership from the age of five does not increase the risk of wheezing and asthma in Japanese children compared to those who have never owned them [40,41].

Inhaling smoke from cigarettes, also known as secondhand smoke (SHS), can cause harm to both smokers and those around them. Exposure to SHS has been linked to higher rates of asthma exacerbations and uncontrolled asthma, even with low levels of exposure [42]. Parents who smoke, particularly fathers, are significantly associated with an increased risk of asthmatic children [43]. Additionally, children exposed to SHS in their homes have a greater risk of stunted growth and a lower height compared with those not exposed. The risk of stunted growth increases with the length of exposure to cigarette smoke [44,45]. This is supported by this study, in which it was found that a higher proportion of stunted asthmatic children was exposed to cigarette smoke compared with stunted children without asthma and those who were non-stunted and asthmatic.

The connection between 25(OH)D levels and asthma in stunted children is a complicated issue that needs further investigation. Various potential factors may contribute to low 25(OH)D levels in stunted asthmatic children, including dietary habits, limited sun exposure, inflammatory response, and the immune system. Vitamin D deficiency during the critical period, from conception to age 2 years, can increase the risk of growth disorders [5]. Vitamin D levels have a significant impact on linear growth and are crucial for normal growth in children [46,47]. Conversely, vitamin D deficiency is correlated with decreased linear and stunting growth [48]. According to research conducted by Van Stuijvenberg et al., children aged 2–5 years in South Africa who are stunted have lower intakes of vitamin D, calcium, riboflavin, and fat compared with children who are not stunted [49]. The inadequate intake of vitamin D is a significant factor related to stunting (adjusted OR = 5.18; 95% CI: 1.03–26.02) [50].

The majority of vitamin D in our bodies is obtained from sunlight, while only about 10% comes from food. When UVB radiation from the sun penetrates the skin, provitamin D3 is converted to previtamin D3 [51]. This is then transformed into calcidiol or 25(OH)D, which can be used to measure vitamin D levels through serum testing [52]. Children with asthma often avoid sun exposure and outdoor activities, preferring to stay indoors and wear warm clothing. As a result, their average daily sun exposure between 10.00 and 15.00 is significantly lower than that of healthy children. This lack of sun exposure is likely the cause of vitamin D deficiency in children with asthma.

Stunted asthmatic children had lower levels of 25(OH)D compared with stunted children without asthma and non-stunted asthmatic children (*p* = 0.042). The median 25(OH)D levels in those groups were below the normal value of <30 ng/mL, according to the Endocrine Society criteria [42]. This finding is consistent with that of Wang et al.’s study, which reported that children with asthma had lower 25(OH)D levels than healthy children [43]. Low levels of 25(OH)D are also associated with a higher prevalence of asthma, hospitalizations, and emergency room visits due to asthma [33]. Malheiro et al. found that children with vitamin D insufficiency had a higher prevalence of severe asthma [53].

This study found that 48.6% of stunted asthmatic children had a deficiency in vitamin D. In Saudi Arabia, 52.1% of children with asthma were reported to have a vitamin D deficiency; a study in Turkey reported similar results of 52.8% [54,55]. Additionally, stunted children were more likely to have deficient 25(OH)D levels than non-stunted children (79.2% vs. 58.3%). A study in Ecuador found lower levels of 25(OH)D in stunted and underweight children aged 6–36 months [5]. It is suggested that this may be due to factors other than asthma, such as stunted growth. It is important to correct any vitamin D deficiencies to prevent a worsening of stunted and asthmatic conditions.

Vitamin D deficiency is thought to be a cause of increased asthma and allergy symptoms [56]. Vitamin D has the ability to modulate the immune system, which is important in the development of asthma. It can inhibit inflammatory signals in various cell types involved in the asthma response and has positive effects on Treg cells and IL-10, both of which suppress Th2 responses [12,13,57]. Studies have shown that perinatal vitamin D deficiency in mice can lead to a reduction in Treg cells that secrete IL-10 and an increase in Th2 cells [11].

Cytokines are involved in immune and inflammatory responses, which contribute to the development of asthma. Asthma patients experience a higher regulation of Th2, which leads to an increase in the production of IL-4 cytokines [58]. IL-4 plays a role in allergic airway disease, causing the differentiation of Th2 lymphocytes, cytokine release, and the induction of isotype class switching B cells to synthesize IgE [15,16,18,59]. This study found that stunted asthmatic children had higher levels of IL-4 compared with stunted children without asthma and non-stunted asthmatic children (*p* = 0.038). This is consistent with previous research conducted by Al-Daghri et al. and Elsaid et al., who also found higher levels of serum IL-4 and IgE in asthma patients compared with non-asthma controls. Additionally, they observed a positive correlation between total IgE and IL-4 levels in asthmatic children [17,58,60]. A study conducted in Syria also reported increased levels of IL-4 in children with asthma compared with controls [17,61]. The levels of IL-4 and IL-13 were also found to be higher in children with recurrent wheezing, with the highest levels observed in those with asthma. Therefore, IL-4 and IL-13 levels may be potential predictors of asthma in children with wheezing [62].

On the other hand, stunted children experience abnormal immune responses, including the increased production of cytokines such as IL-4 and total IgE compared with well-nourished children [23,25]. This study is the first to report IL-4 levels in stunted asthmatic children. IL-4 levels in stunted asthmatic children were significantly higher than those in controls, as they were influenced by both stunted growth and asthma. However, until now, there has been no research on the effect of an improved nutritional status on reducing IL-4 levels in stunted children. A recent in vitro study performed by Rodriguez et al. reported that CD4+ cells incubated with leptin exhibited increases in IL-2 and IFN-γ production and decreases in IL-4 and IL-10 production [63]. Based on research performed by Rodriguez et al., improving the nutritional status is expected to reduce IL-4 levels and improve asthma outcomes in stunted children [63].

Interleukin-10 is a type of cytokine that has a role in allergic diseases by influencing various cell functions such as the activation of Th2 cells, the function of mast cells and eosinophils, and the ratio of IgG to IgE. IL-10 has been considered in the immune response as a promotive and suppressive cytokine. Due to its anti-inflammatory role, IL-10 has also been reported to enhance immune events via immunoglobulin production by B cells, the cytotoxicity of NK cells and CD8+ T cells, and the proliferation of thymocytes [64]. Additionally, IL-10 hinders the function of APC, which includes the maturation of dendritic cells and the expression of MHC class II and co-stimulatory molecules [65]. Studies have shown that there is a relationship between IL-10 levels and the severity of allergic disease and asthma, with lower levels of IL-10 associated with more severe disease. In stunted and asthmatic cases, IL-10 levels tend to be lower than those in control groups. These findings confirm the results of Zheng et al.’s study, which revealed that atopic asthma patients had lower levels of IL-10 compared with controls [66]. Yuksell et al. also found that before administering montelukast treatment, serum IL-10 levels in children with asthma significantly decreased compared with controls [67]. Tsai et al. discovered that mean serum IL-10 levels in asthmatic individuals not experiencing attack and those with acute exacerbations were lower than those in controls [68]. The non-significant difference in IL-10 levels may be attributed to the conflicting results of previous studies on IL-10 levels in children with asthma (most studies reported low IL-10 levels; a few reported high IL-10 levels), as well as the contradictory results between low IL-10 levels in children with asthma and high IL-10 levels in stunted children (Figure 1).

This study examined the balance between the levels of IL-4 and IL-10 in the body, as measured by the IL-4/IL-10 ratio. The researchers found that stunted asthmatic children had higher IL-4/IL-10 ratios compared with stunted children without asthma and non-stunted children with asthma. A high IL-4/IL-10 ratio can be caused by high IL-4 levels, low IL-10 levels, or both. In stunted asthmatic children, the ratio was high due to increased IL-4 and low IL-10 levels. A higher IL-4/IL-10 ratio suggests a shift towards a pro-inflammatory immune response, as IL-4 is linked to a pro-inflammatory Th2 response, while IL-10 is anti-inflammatory.

CD23+ is a low-affinity receptor for IgE found mainly on the surface of B lymphocytes, underlying a pivotal mechanism of IgE homeostasis in humans. Its main function is to regulate IgE responses and play a crucial role in facilitating the presentation of allergens to T cells [19,69]. This process activates allergen-specific T cells and triggers the secretion of Th2-driving cytokines. The presence of IgE can lead to an upregulation of CD23+ in B cells. The density of CD23+ molecules on the surface of B cells affects allergen-specific IgE levels, which, in turn, affect allergen uptake and subsequent T cell proliferation and activation. The number of CD23+ molecules on the surface of antigen-presenting cells (APCs) correlates with IgE-facilitated allergen uptake, which is dependent on allergen-specific IgE. These findings have implications for the treatment of allergic diseases [19,69,70]. The activation of T cells was strongest when APCs expressed high levels of CD23+. There was also a significant correlation between CD23+ expression on B cells and T cell activation. Patients with allergies exhibited increased surface density expression of CD23+ on their B cells due to high levels of allergen-specific IgE. This, in turn, led to increased IgE-facilitated allergen presentation and the activation of allergen-specific T cells.

This study revealed that individuals who were stunted and had asthma had higher CD23+ expression (measured in both percentage and MFI) compared with the control groups. However, the result was not significant. This finding supports previous research performed by Chary et al. and Aberle et al., who reported that children with extrinsic (allergic) asthma exhibited a significantly higher percentage of CD23+ in their B cells than children with intrinsic asthma or healthy children [71,72]. Hagel et al. also found that stunted children exhibited an increased proportion of B cells with low-affinity IgE receptors (CD23+) [26]. Nonetheless, this study is the first to report on CD23+ expression in children who are stunted and asthmatic.

The CD23+ expression results were not significant, which could be explained by the research of Oettgen et al., who reported that IgE regulates its receptors, namely, FcϵRI and CD23+ [73]. Cells cultured with IgE expressed higher levels of FcϵRI and CD23+. B cells and mast cells from IgE -/- mice exhibited decreased levels of FcϵRI and CD23+, but these levels returned to normal after intravenous IgE infusion. The cross-linking of membrane-bound CD23+ and mIgE by an allergen–serum IgE complex results in the suppression of IgE synthesis. Findings in mouse studies suggested that the binding of IgE to CD23+ expressed on B cells, rather than to FceRI present on basophils or mast cells, regulates a pool of free serum IgE [74]. Thus, CD23+ expression is affected by specific IgE levels. This study did not examine specific IgE levels, which affect CD23+ expression. Consequently, it is unknown how IgE would have affected CD23+ expression between subject groups. This may explain why CD23+ expression was not significant in this study (Figure 1).

The findings of this study have some limitations: (1) There were few research participants because the prevalence of asthma in children is very low. (2) Non-stunted non-asthmatic children were not involved because this study was part of the vitamin D Academic Leadership Grant research with the title ‘Relationship between vitamin D and contributing factors to the incidence and comorbidities of stunting in toddlers aged 0–59 months in Bandung Regency’, which only involved stunted children. In addition, this study was carried out during the ongoing SARS-CoV-2 pandemic, where subject recruitment was very limited and it was difficult to involve healthy children. (3) There are confounding variables that could affect the levels of independent variables (such as vitamin D, IL-4, IL-10, and CD23+ expression) or dependent variables (such as bronchial asthma) that could not be measured in this study. The confounding variables that affect vitamin D levels include sun exposure and food intake, while allergen exposure and genetic variation can influence IL-4 and IL-10 levels, as well as CD23+ expression. (4) We did not conduct a serum-specific IgE examination due to the complexity of the process, which requires various types of allergens and can be quite expensive. Long-term follow-up studies with larger participant cohorts are necessary to observe changes in vitamin D, IL-4, IL-10, and CD23+ levels and the development of asthma to explain and prove the effect of these variables on the incidence of bronchial asthma.

## 5. Conclusions

In stunted children, there is an association between low levels of vitamin D and high levels of interleukin-4 with bronchial asthma. Pediatricians should recommend checking 25(OH)D levels in both stunted asthma and non-stunted asthmatic children to determine their vitamin D status. If low levels of vitamin D are detected, a simple and cost-effective way to increase 25(OH)D levels is to encourage children to regularly sunbathe between 10:00 am and 02:00 pm Pediatricians also recommend examining IL-4 levels in stunted children with low vitamin D levels and a family history of atopy disease to determine their risk of developing bronchial asthma. Further research is needed to determine the impact of vitamin D supplementation on improving bronchial asthma outcomes in stunted children. Long-term follow-up studies are also needed to determine changes in vitamin D, IL-4, and CD23+ levels and their relationship with the development of bronchial asthma in stunted children.

## Figures and Tables

**Figure 1 biomedicines-11-02542-f001:**
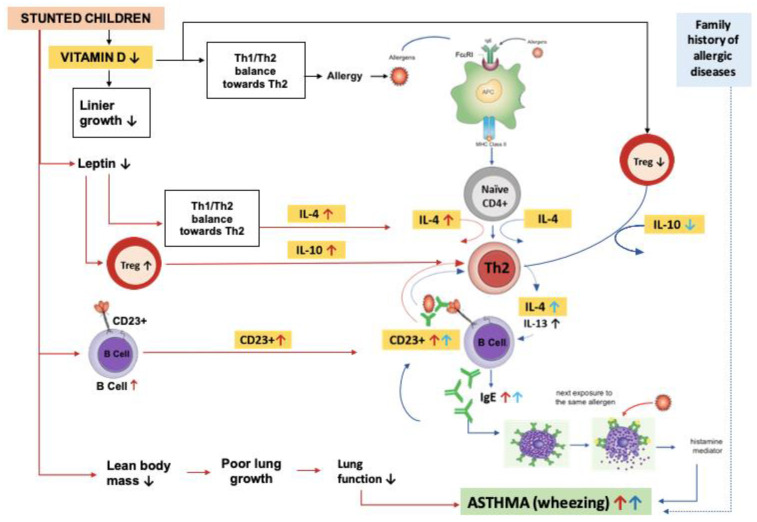
Possible mechanism of asthma in stunted children. Stunted children have lower levels of vitamin D, which can lead to a decrease in linear growth. Low levels of vitamin D can also cause an imbalance in Th1–Th2 cells, which can lead to allergic diseases such as asthma. The development of asthma begins with exposure to allergens, which are captured by antigen presenting cells (APCs) and presented to naive CD4+ cells. With the help of IL-4, these cells become Th2 cells that produce IL-4 and IL-13. B lymphocytes then become plasma cells that produce IgE and attach to mast cells. Further exposure to allergens can trigger the release of histamine and lead to bronchoconstriction, causing wheezing and other asthma symptoms. When vitamin D levels are low, the number of Treg cells decreases, which can lead to an increase in Th2 activity and IL-4 production. This can result in an increase in IgE, which can further stabilize CD23+ B cells. B cells can then take up allergen–IgE complexes and present the allergen to Th2 cells, leading to an increase in the Th2 response. In stunted children, a decrease in lean body mass can cause impaired lung growth and decreased lung function. Low leptin levels can cause an imbalance in Th1–Th2 cells, leading to an increase in IL-4 production. In addition, it can cause an increase in Treg cells and IL-10, different from asthmatic children. B cells also increase, causing an increase in CD23+ levels. All of these factors can contribute to the development of asthma in stunted children. (red arrows: stunting paths, blue arrows: asthma paths).

**Figure 2 biomedicines-11-02542-f002:**
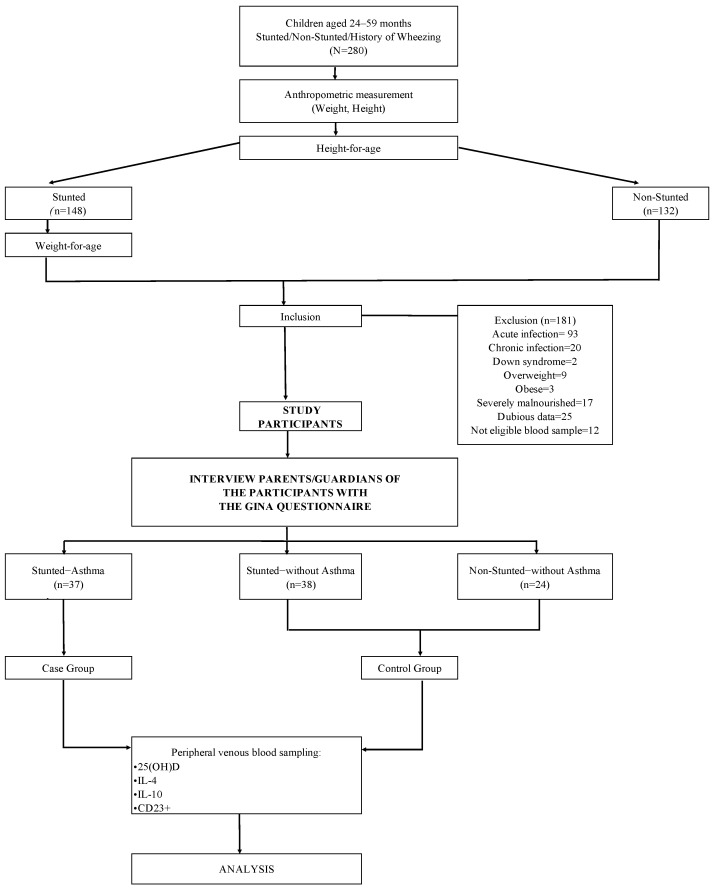
Flow chart of the selection process for study participants.

**Figure 3 biomedicines-11-02542-f003:**
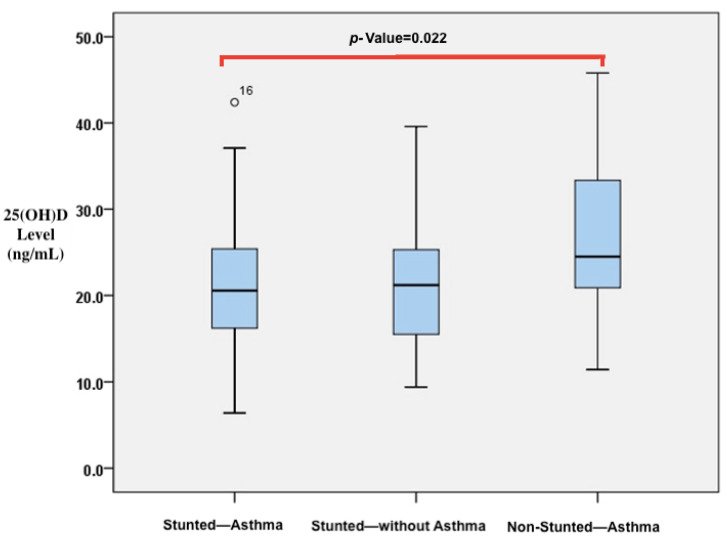
Differences in 25(OH)D levels between stunted asthmatic children, stunted children without asthma, and non-stunted asthmatic children.

**Figure 4 biomedicines-11-02542-f004:**
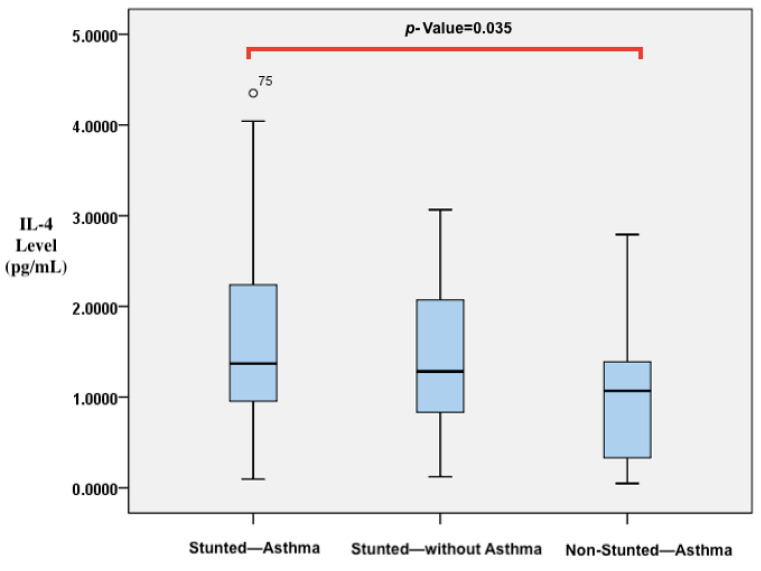
Differences in IL-4 levels between stunted asthmatic children, stunted children without asthma, and non-stunted asthmatic children.

**Table 1 biomedicines-11-02542-t001:** Characteristics of the study participants ^a^.

Characteristic	Stunted Asthmatic (*n* = 37)	Stunted without Asthma (*n* = 38)	Non-Stunted Asthmatic (*n* = 24)	*p*-Value ^b^
Sex				
Male	22 (59.5%)	17 (44.7%)	15 (62.5%)	0.294
Female	15 (40.5%)	21 (55.3%)	9 (37.5%)	
Age (months) Mean (SD)	41.4 (9.9)	39.7(9.6)	45.5 (10.3)	0.082
Weight for height				
Normal	36 (97.3%)	36 (94.7%)	24 (100%)	0.364
Wasted	1 (2.7%)	2 (5.3%)	0	
Asthma severity				
Intermittent	26 (70.3%)	-	16 (66.7%)	0.767
Mild Persistent	11 (29.7%)	-	8 (33.3%)	
Allergen exposure				
Cat ownership when 0–12 months old	4 (10.8%)	2 (5.3%)	3 (12.5%)	0.565
Cat ownership in the past 12 months	4 (10.8%)	3 (7.9%)	3 (12.5%)	0.828
Dog ownership when 0–12 months old	0	0	1	0.206
Dog ownership in the past 12 month	0	0	0	-
History of allergic diseases				
Father with atopy	10 (27.0%)	2 (5.3%)	5 (20.8%)	0.038
Mother with atopy	20 (54.1%)	5 (13.2%)	11 (45.8%)	0.001
Both father and mother with atopy	5 (13.5%)	1 (1.6%)	2 (8.3%)	0.224
Siblings with atopy	12 (32.4%)	3 (7.9%)	4 (16.7%)	0.025
Smoking exposure				
Yes	33 (89.2%)	32 (84.2%)	18 (75.0%)	0.338
No	4 (10.8%)	6 (15.8%)	6 (25.0%)	
Delivery				
Vaginal	33 (89.2%)	31 (81.6%)	19 (79.2%)	0.519
Caesarean section	4 (10.8%)	7 (18.4%)	2 (20.8%)	
Lactation				
Exclusive Breastfeeding	5 (13.5%)	8 (21.1%)	17 (70.8%)	<0.001
Breastfeeding and Formula milk	32 (86.5%)	30 (78.9%)	7 (29.2%)	
Formula milk	-	-	-	
Birth weight				
<2500 g	5 (13.5%)	3 (7.9%)	2 (8.3%)	0.684
≥2500 g	32 (86.5%)	35 (92.1%)	22 (91.7%)	
Gestational age				
Preterm	2 (5.4%)	3 (7.9%)	1 (4.2%)	0.807
Term	34 (91.9%)	32 (84.2%)	21 (87.5%)	
Post-term	1 (2.7%)	3 (7.9%)	2 (8.3%)	
Leukocyte count (×10^3^ cells/mm^3^) Mean (SD)	10.50 (2.98)	9.78 (2.26)	10.97 (2.94)	0.288

Note: ^a^ Analysis of variance was performed for normally distributed data, and the Kruskal–Wallis test was performed for non-normally distributed data. For categorical and nominal data, chi-squared tests were used. ^b^
*p* < 0.05 was considered significant.

**Table 2 biomedicines-11-02542-t002:** Profile of study participants’ serum 25(OH)D levels.

	Group	Stunted Asthmatic (*n* = 37)	Stunted without Asthma (*n* = 38)	Non-Stunted Asthmatic (*n* = 24)	*p*-Values
Category ^a^					
Deficiency ≤20 ng/mL		18 (48.6%)	17 (44.7%)	5 (20.8%)	0.171 ^b^
Insufficiency 21–29 ng/mL		15 (40.5%)	14 (36.8%)	12 (50%)	
Sufficient ≥30 ng/mL		4 (10.8%)	7 (18.4%)	7 (29.2%)	

Note: ^a^ Classification based on the Endocrine Society criteria. ^b^ Chi-squared test.

**Table 3 biomedicines-11-02542-t003:** Comparison of 25(OH)D, IL-4, and IL-10 levels and CD23+ expression in the stunted asthmatic children, stunted children without asthma, and non-stunted asthmatic children ^a^.

	Group	Stunted Asthmatic (*n* = 37)	Stunted without Asthma (*n* = 38)	Non-Stunted Asthmatic (*n* = 24)	*p*-Value ^b^
Variable					
25(OH)D level (ng/mL)		20.55 (16.18–25.55) ^c^	21.2 (15.45–25.4) ^c^	24.50 (20.90–34.02) ^d^	0.042
IL-4 level (pg/mL)		1.41 (0.95–2.40) ^c^	1.34 (0.83–2.17) ^c^	1.09 (0.30–1.58) ^d^	0.038
IL-10 level (pg/mL)		0.76 (0.26–1.68) ^c^	0.87 (0.26–1.81) ^c^	1.07 (0.18–1.64) ^c^	0.956
Ratio IL-4/IL-10		1.963 (0.744–5.610) ^c^	1.716 (0.848–6.353) ^c^	1.221 (0.423–2.084) ^c^	0.099
Expression of CD23+/Percentage		39.1 (30.65–50.55) ^c^	38.75 (32.02–50.45) ^c^	37.05 (12.02–48.78) ^c^	0.393
Expression of CD23+/MFI		982 (747–1367) ^c^	939 (770–1272) ^c^	763 (611–1228) ^c^	0.091

Note: ^a^ median (inter quartile range). ^b^
*p* < 0.05 was considered significant based on the Kruskal–Wallis test. ^c^ the same letter within rows indicates that there was no significant difference (*p* < 0.05). ^d^ the different letter within rows indicates that there was significant difference (*p* < 0.05).

## Data Availability

The data presented in this study are available on request from the corresponding author.

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
