# Peer review of "The Association between Vitamin D, Interleukin-4, and Interleukin-10 Levels and CD23+ Expression with Bronchial Asthma in Stunted Children"

_biomedicines, 2023, doi:10.3390/biomedicines11092542_

Round 1

Reviewer 1 Report

With interest, I read the manuscript biomedicines-2573245. Despite some evident drawbacks, this work might have some potential for a publication in Biomedicines.

Specific comments (no special order):

1.      Abstract needs to be rewritten. It contains some unnecessary details, e.g. “in Bandung”. It also does not have a flow and the order of the sentences is not correct. Please, read once again and critically correct.

2.      The same applies to the whole manuscript, especially Introduction and Discussion where the overall flow needs to be improved.

3.      It would be good if any colleague having more experience with the English language critically read this work. Such language expert could also help to eliminate places like lines 85-91 where “facilitate” is repeated three times, including “It facilitates IgE-facilitated allergen presentation …”…

4.      The relationships between FcεRI, FcεRII and serum IgE are quite well described. But thei are even more complex. Please, refer to PMID: 22909159.

5.      The same applies to the part on adipokines/leptin. Still, please, refer to PMID: 34948451 and 36613991.

6.      Overall, having some knowledge in the field, I can understand how the Authors combined vitamin D, IL-4, IL-10 levels, and CD23+ expression with bronchial asthma but for any less experienced Reader the story can be too thin. Please, try to work on making your hypothesis presented in the Introduction even more convincing.

7.      Why the study does not include a group of non-stunted non-asthmatic children? Explain in the limitations, please.

8.      The laboratory methodology must be described in more detail, especially FACS. In all cases the names of the kits and reagents with manufacturer, (state), city, country should be added, too.

9.      Were the samples used for FACS analyses fresh?

10.   How were the other samples stored before analyses?

11.   Lines 169-177. Overall, the content of this chapter does not make sense. Please, consult the statistician.

12.   Table 1. How are continuous data presented? “For categorical.”? Exactly, what test have been used for categorical data?

13.   Anyway, you should better make three columns for p-values and present all pairwise comparisons between the groups already here. Table 2 should be deleted.

14.   Add pairwise p-values to Table 3.

15.   The Results section should be divided into subchapters.

16.   Multiple fonts are used throughout the manuscript. Please, unify.

17.   Table 4. Add all pairwise p-values to Table 4 instead of KW p-value.

18.   Figure 1. How many were excluded (“Exclusion”)?

19.   Figures 2 and 3. Do not use commas for decimals. Add pairwise comparison p-values and specific testing methods,

20.   But in fact, please, remove figures 2 and 3. They only duplicated the content of Table 4.

21.   Btw., please, calculate correlations between the parameters given in Table 4.

22.   How was Table 5 created if of non-stunted non-asthmatic children were not included in the study? Besides, what is the dependent variable here, asthma? What do you mean, “in stunted children against non-stunted.” -  is it an independent variable as well. Discuss with the statistician or the best remove Table 5 and the respective part of the main text completely as it does not make any statistical sense or at least not the sense you are trying to report.

23.   Nutrition affects allergies and asthma risk also prenatally (PMID: 33668787). Please, address in the Discussion.

24.   Finally, IL-10 plays an important role in mast cells as well (PMID: 34067047). Please, address in the Discussion.

Moderate editing of English language required. See also my other comments.

Author Response

Dear Reviewer 1,

Thank you for taking the time to review our manuscript. We have incorporated your suggestions and feedback to improve it. If you have any further concerns, please let us know.

Best Regards,

  1. Abstract needs to be rewritten. It contains some unnecessary details, e.g. “in Bandung”. It also does not have a flow and the order of the sentences is not correct. Please, read once again and critically correct.

Answer: 

The abstract has been rewritten, removing unnecessary details and improving sentence flow and order. Suggestions and corrections are welcome.

  1. The same applies to the whole manuscript, especially Introduction and Discussion where the overall flow needs to be improved. 

Answer:

The entire manuscript, including the Introduction and Discussion, has been revised. Kindly provide any suggestions or corrections.

  1. It would be good if any colleague having more experience with the English language critically read this work. Such language expert could also help to eliminate places like lines 85-91 where “facilitate” is repeated three times, including “It facilitates IgE-facilitated allergen presentation …”… 

Answer:

Thank you for your feedback. An English language expert (MDPI English editing) has corrected the language in my manuscript.

  1. The relationships between FcεRI, FcεRII and serum IgE are quite well described. But thei are even more complex. Please, refer to PMID: 22909159. 

Answer: 

Thank you for your reference. I have added to the discussion on the mechanism of FcεRII/CD23+ and IgE.

  1. The same applies to the part on adipokines/leptin. Still, please, refer to PMID: 34948451 and 36613991. 

Answer: 

Thank you for providing your reference. I have included the information regarding leptin in the introduction.

  1. Overall, having some knowledge in the field, I can understand how the Authors combined vitamin D, IL-4, IL-10 levels, and CD23+ expression with bronchial asthma but for any less experienced Reader the story can be too thin. Please, try to work on making your hypothesis presented in the Introduction even more convincing. 

Answer:

I added a hypothesis in the introduction to clarify the association between vitamin D, IL-4, and IL-10 levels, and CD23+ expression with asthma in stunted children. This can be seen in Figure 1.

  1. Why the study does not include a group of non-stunted non-asthmatic children? Explain in the limitations, please. 

Answer:

Non-stunted non-asthmatic children were not involved because this study was part of the vitamin D Academic Leadership Grant research with the title ‘Relationship between vitamin D and contributing factors to the incidence and comorbidities of stunting in toddlers aged 0-59 months in Bandung Regency’, which only involved stunted children. In addition, this study was carried out during the ongoing SARS-CoV-2 pandemic, where subject recruitment was very limited and it was difficult to involve healthy children. We have acknowledged this limitation in our research and included it in the manuscript.

  1. The laboratory methodology must be described in more detail, especially FACS. In all cases the names of the kits and reagents with manufacturer, (state), city, country should be added, too. 

Answer: 

Thank you for your feedback. I have already provided detailed information about the laboratory methodology in the manuscripts.

  1. Were the samples used for FACS analyses fresh? 

Answer: 

Yes, we used fresh blood samples for FACS analyses. I have already added additional information about blood sampling in the methodology section.

  1. How were the other samples stored before analyses? 

Answer: 

The other serum samples were stored at -80°C for examination of vitamin D, IL-4, and IL-10 levels and this information has already included in the manuscripts.

  1. Lines 169-177. Overall, the content of this chapter does not make sense. Please, consult the statistician. 

Answer:

I have already consulted the stastistician and have revised about the content in data analysis.

  1. Table 1. How are continuous data presented? “

Answer: 

The age and leukocyte count are presented as continuous data in mean (SD) format.

  • For categorical.”? Exactly, what test have been used for categorical data?

Answer: 

“For categorical and nominal data, chi-squared tests were used.” The sentence was incomplete, so I have made revisions to the manuscript.

  1. Anyway, you should better make three columns for p-values and present all pairwise comparisons between the groups already here. 

Answer:  

After consulting a statistician, I learned that if analysis results are not significant, there is no need to conduct pairwise comparisons between groups again.

Table 2 should be deleted.

 Answer: 

Table 2 has already deleted.

  1. Add pairwise p-values to Table 3. 

Answer: 

Thank you for your feedback. I have added the p-values as requested.

  1. The Results section should be divided into subchapters. 

Answer: 

Thank you for providing your feedback. We would like to inform you that we have divided the Results section into subchapters, as per your suggestion.

  1. Multiple fonts are used throughout the manuscript. Please, unify. 

Answer: 

I apologize for using multiple fonts in the manuscript. I have already unified them

  1. Table 4. Add all pairwise p-values to Table 4 instead of KW p-value.

Answer: 

I have already added all pairwise p-values to Table 4.

  1. Figure 1. How many were excluded (“Exclusion”)? 

Answer: 

There were 181 participants excluded. The number of exclusions has been added to Figure 2.

  1. Figures 2 and 3. Do not use commas for decimals. Add pairwise comparison p-values and specific testing methods,

Answer : 

Figures 3 and 4 replace Figures 2 and 3. The comma is now a full stop. Significant pairwise comparison p-values were included.

  1. But in fact, please, remove figures 2 and 3. They only duplicated the content of Table 4.

Answer: 

Figures 2 and 3 (which change to Figures 3 and 4) show the results of post hoc analysis of significant levels of vitamin D and IL-4 between groups, so they are not the same as Table 4. The detailed explanation is as follows:

“In post hoc analysis, it was found that significant differences existed between the group of stunted–asthmatic children and non-stunted–asthmatic children. Vitamin D levels in stunted–asthmatic children were lower than non-stunted–asthmatic children (p=0.022). However, IL-4 levels in stunted–asthmatic children were higher than non-stunted–asthmatic children (p=0.035). The results of the analysis are reflected in the p-values in Figure 3 and Figure 4.”

Could you kindly clarify if there is still a need to remove figures 2 and 3? Thank you.

  1. , please, calculate correlations between the parameters given in Table 4.

Answer: 

This study does not calculate the correlation between parameters in Table 4 because it is not the aim of this study. The purpose of this study was to analyze the association between the independent variables (vitamin D, IL-4, IL-10 , and CD23+) and the dependent variable (bronchial asthma).

  1. How was Table 5 created if of non-stunted non-asthmatic children were not included in the study? 

Answer:

Table 5 shows the predictive model in the multiple logistic regression analysis. The purpose of creating this table is to assist in understanding and predicting the association between predictor/independent variables ((vitamin D, IL-4, IL-10, CD23+) simultaneously and the response/dependent variable (bronchial asthma) by including confounding factors in the multivariable analysis. 

Considering that this study did not involve children who were not stunted non asthmatic children, then from the results of the bivariable analysis there were significant differences between the groups of stunted-asthmatic vs. non-stunted-asthmatic children, a predictive model was created by comparing stunted-asthmatic children vs. non-stunted- asthmatic children (differentiating factor is stunted vs. non-stunted).

  • Besides, what is the dependent variable here, asthma?

Answer: The dependent variable in this study is bronchial asthma.

  • What do you mean, “in stunted children against non-stunted.” -  is it an independent variable as well. 

Answer:

It means "stunted-asthma children versus non-stunted-asthma children". It is not an independent variable but a case group versus a control group.

After I discuss it again with the statistician and consider your suggestions, it is better to just delete Table 5. Please provide suggestions and corrections.

  • Discuss with the statistician or the best remove Table 5 and the respective part of the main text completely as it does not make any statistical sense or at least not the sense you are trying to report.

Answer:

After I discuss it again with the statistician and consider your suggestions, it is better to delete Table 5 and all parts of the main text related to Table 5. Please provide suggestions and corrections.

  1. Nutrition affects allergies and asthma risk also prenatally (PMID: 33668787). Please, address in the Discussion. 

Answer: 

Thank you for your reference. I have added some points from that journal to support the data on new variables in basic characteristics, specifically lactation.

  1. Finally, IL-10 plays an important role in mast cells as well (PMID: 34067047). Please, address in the Discussion. 

Answer : 

Thank you for your reference. I have already added a discussion about the role of IL-10 in the immune system.

Authors

Reviewer 2 Report

This paper entitled “The Association between Vitamin D, Interleukin-4, and Interleukin-10 Levels and CD23+ Expression with Bronchial Asthma in Stunted Children” examined the role of vitamin D, IL-4, IL-10, and CD23+ in stunted children aged 24-59 months in Bandung.

The topic is of interest to the readers of the Journal. I consider the topic original, and the study has the potentiality of being shared with the scientific community. The References are updated to the most recent research. The manuscript is well articulated, the arguments are presented in a manner consistent with the hypotheses formulated, and the literature review is satisfactory and up-to-date. 

Main concerns:

ABSTRACT: The authors should start with a short intro that better highlights their work.

METHODS: 

- Experimental procedures should be better defined

- More information should be provided about the participants’ characteristics

- What were inclusion and exclusion criteria? 

DISCUSSION: The discussion should start with a first paragraph describing the main aims and then the main results.

Kind regards

Author Response

Dear Reviewer 2,

Thank you for taking the time to review our manuscript. We have incorporated your suggestions and feedback to improve it. If you have any further concerns, please let us know.

Best Regards,

Authors

REVIEWER 2

Main concerns:

ABSTRACT:

  • The authors should start with a short intro that better highlights their work. 

Answer:

The author has added a brief introduction to the abstract highlighting their work and requested suggestions and corrections.

METHODS: 

  • Experimental procedures should be better defined 

Answer:

I have made improvements to the experimental procedures. I welcome any suggestions or corrections you may have.

  • More information should be provided about the participants’ characteristics

Answer:

Additional details on participant characteristics have been included in the manuscript.

  • What were inclusion and exclusion criteria? 

Answer:

The manuscript includes both inclusion and exclusion criteria, which are listed below:

Inclusion criteria are as follows: (1) stunted–asthma if stunted children (HFA <-2 SD with WFA >-3 SD to 2 SD according to the WHO child growth standard/WHOCGS curve) with bronchial asthma, (2) stunted–without asthma if stunted children without asthma, and (3) non-stunted–asthma if children with normal height and weight according to WHOCGS curve with bronchial asthma.  

Participants who met the following exclusion criteria were not included in the study: (1) those with major congenital abnormalities or certain syndromes such as Down syndrome; (2) stunted children with overweight, obese, and severely malnourished (weight for age <-3 SD); (3) children with acute and chronic infections; (4) those who have received vitamin D supplementation; and (5) children with bronchial asthma who have received steroid inhalation therapy for ≥12 weeks with a dose of 200 micrograms or higher.

DISCUSSION:

  • The discussion should start with a first paragraph describing the main aims and then the main results. 

Answer:

The first paragraph of the discussion now explains the main objectives followed by the main results.

Reviewer 3 Report

Dear Authors,

I have read your manuscript and I send you my comments:

1 Minor comments:

Results line 1: I think that there is an error "....had stunted-asthma, 38 had stunted-asthma.."

Table 4: P value is referred to?

Please change gender to sex

2) Major comments:

- Please add data of lactation

- Patients' number is very low please add patients

- power calculation is missing please add it

- The group non-stunded without asthma is missing please add it

- please add immunochemistry data of the subtypes of T lymphocites for each group

- Data of IL-4 and  vitamin D have been documented in the selected population

- No data regarding the patients with asthma have been reported, please adda data of spirometry

- Conclusion: "Additionally, vitamin D supplementation can be used to improve bronchial asthma outcomes" Please delete it, you did not evaluate the effects of vitamin D administration in these patients

Author Response

Dear Reviewer 3,

Thank you for taking the time to review our manuscript. We have incorporated your suggestions and feedback to improve it. If you have any further concerns, please let us know.

Best Regards,

Authors

REVIEWER 3

1 Minor comments:

  • Results line 1: I think that there is an error "....had stunted-asthma, 38 had stunted-asthma.."

Answer:

There was an error in the original text. The correct sentence is: "37 had stunted-asthma, 38 had stunted-without asthma."

  • Table 4: P value is referred to?

Answer:

The p-value in Table 4 indicates if there is a significant difference between the stunted-asthma, stunted-without asthma, and non-stunted-asthma groups for the variables 25(OH)D, IL-4, IL-10, IL-4/IL-4 ratio, IL-10, and CD23, using the Kruskal-Wallis test.

  • Please change gender to sex

Answer:

I have already change gender to sex. 

2) Major comments:

  • Please add data of lactation

Answer:

We have added the lactation data as a new variable in the Table 1. Characteristics of the study participants

Characteristic

Stunted-Asthma

(n= 37)

Stunted-

without Asthma

(n= 38)

Non-Stunted-Asthma

(n= 24)

p-valueb

Lactation

·       Exclusive Breastfeeding

·       Breastfeeding + Formula milk

·       Formula milk 

5 (13.5%)

32 (86.5%)

-

8 (21.1%)

30 (78.9%)

-

17 (70.8%)

7 (29.2%)

-

<0.001

  • Patients' number is very low please add patients 

Answer:

Although the number of asthmatic children who are stunted or non-stunted is very low, we have used statistical calculations to determine that a minimum sample size of 20 is necessary for each group. In Indonesia, the prevalence of asthma in children aged 1-4 years is 1.6%, while in those aged 5-14 years, it is 1.9%. Due to the rarity of cases, it can be challenging to find asthmatic children who are stunted or non-stunted. Our coverage area currently includes 1 district in Bandung with 62 primary care facilities. To increase the number of patients, a larger coverage area is required, which will be implemented in a multicenter manner. I have included this information in the section discussing the limitations of the research.

  • power calculation is missing please add it

Answer:

The power calculation is included in the method section.

  • The group non-stunded without asthma is missing please add it

Answer:

Non-stunted non-asthmatic children were not involved because this study was part of the vitamin D Academic Leadership Grant research with the title ‘Relationship between vitamin D and contributing factors to the incidence and comorbidities of stunting in toddlers aged 0-59 months in Bandung Regency’, which only involved stunted children. In addition, this study was carried out during the ongoing SARS-CoV-2 pandemic, where subject recruitment was very limited and it was difficult to involve healthy children. We have acknowledged this limitation in our research and included it in the manuscript.

  • please add immunochemistry data of the subtypes of T lymphocites for each group

Answer:

This study does not include immunochemical data on T lymphocyte subtypes for each group as it was not the aim of the study.

  • Data of IL-4 and vitamin D have been documented in the selected population

  • No data regarding the patients with asthma have been reported, please adda data of spirometry

Answer:

This study focused on children between the ages of 24 to 59 months. In children under the age of 5, asthma diagnosis is determined solely by clinical presentation and physical examination, following GINA guidelines. This is due to the inability to perform spirometry tests in children of this age.

  • Conclusion: "Additionally, vitamin D supplementation can be used to improve bronchial asthma outcomes" Please delete it, you did not evaluate the effects of vitamin D administration in these patients

Answer:

Thank you for your feedback. I have already deleted it.

Round 2

Reviewer 1 Report

My comments have been addressed well. Thank you.

Reviewer 3 Report

no comments

none